# GUIDING PHYSICAL INTUITION WITH NEURAL STETHOSCOPES

## ABSTRACT

Model interpretability and systematic, targeted model adaptation present central challenges in deep learning. In the domain of intuitive physics, we study the task of visually predicting stability of block towers with the goal of understanding and influencing the model's reasoning. Our contributions are two-fold. Firstly, we introduce *neural stethoscopes* as a framework for quantifying the degree of importance of specific factors of influence in deep networks as well as for actively promoting and suppressing information as appropriate. In doing so, we unify concepts from multitask learning as well as training with auxiliary and adversarial losses. Secondly, we deploy the stethoscope framework to provide an in-depth analysis of a state-of-the-art deep neural network for stability prediction, specifically examining its physical reasoning. We show that the baseline model is susceptible to being misled by incorrect visual cues. This leads to a performance breakdown to the level of random guessing when training on scenarios where visual cues are inversely correlated with stability. Using stethoscopes to promote meaningful feature extraction increases performance from 51% to 90% prediction accuracy. Conversely, training on an easy dataset where visual cues are positively correlated with stability, the baseline model learns a bias leading to poor performance on a harder dataset. Using an adversarial stethoscope, the network is successfully de-biased, leading to a performance increase from 66% to 88%.

## 1 INTRODUCTION

Intuitive physics in the deep learning community describes physical understanding acquired by neural networks in a data-driven as opposed to a rule-based manner: With an increasing amount of training examples, we expect an algorithm to develop a better understanding of its (physical) environment, especially when the task it is trained on is inherently linked to the physical rules governing the scene. However, what type of understanding the network develops highly depends on the types of scenarios it is confronted with and the task it is trying to solve. Furthermore, it depends on the network architecture, on regularisation techniques, on the training procedure, *etc.* As a result, in contrast to a rule-based approach, it is often hard to assess what form of physical understanding a neural network has developed. We are specifically interested in whether the network uses visual cues as shortcuts which reflect correlations in the dataset but are incommensurate with the underlying laws of physics the network was intended to learn.

In this paper, we specifically focus on stability prediction of block towers, a task which has gained interest in both the deep learning (Lerer et al., 2016; Wu et al., 2017; Groth et al., 2018) and the robotics community in recent years (Li et al., 2017a;b). Images of towers of blocks stacked on top of each other are shown to a neural network. Its task is to predict whether the tower will fall over or not resulting in a binary classification problem. End-to-end learning approaches as well as simulation-based approaches achieve super-human performance on a real dataset (Lerer et al., 2016; Wu et al., 2017; Groth et al., 2018). However, with investigation of trained deep learning models limited to occlusion-based attention analyses (Lerer et al., 2016; Groth et al., 2018), it is not clear to what extent neural networks trained on this task take into account physical principles such as centre-of-mass or whether they follow visual cues instead. To this end, we introduce a variation of the ShapeStacks dataset presented by Groth et al. (2018) which facilitates the analysis of the effects of visual cues on the learning process.

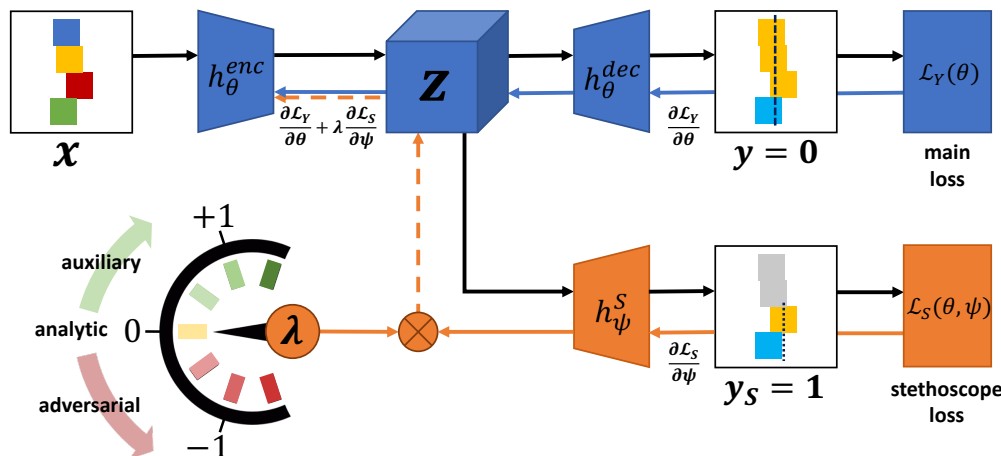

Figure 1: The stethoscope framework. The main network (blue), comprised of an encoder and a decoder, is trained for global stability prediction of block towers. The stethoscope (orange), a two layered perceptron, is trained to predict a nuisance parameter (local stability) where the input is $Z$, a learned feature from an arbitrary layer of the main network. The stethoscope loss is back-propagated with weighting factor $\lambda$ to the main network. The value of $\lambda$ determines whether the stethoscope operates in analytic ($\lambda = 0$), auxiliary ($\lambda > 0$) or adversarial manner ($\lambda < 0$).

Motivated by the need for an effective tool to understand and guide the physical reasoning of the neural network and inspired by prior research in interpretability, multi-task learning and adversarial training, we present *neural stethoscopes* as a unified framework for the interrogation and perturbation of task-specific information at any layer. A stethoscope can be deployed in a purely *analytic* fashion whereby a question is posed via a stethoscope loss which is not propagated back into the main network. It can also be used to promote or suppress specific information by deploying either an auxiliary or an adversarial training mechanism. The concept is illustrated in Figure 1. We demonstrate that deploying an auxiliary stethoscope can be used to promote information conducive to the main task improving overall network performance. Conversely, we show that an adversarial stethoscope can mitigate a specific bias by effectively suppressing information. Moreover, the main network does not need to be changed in order to apply a neural stethoscope.

In this work, we present two contributions: (1) An in-depth analysis of the state-of-the-art approach for intuitive stability prediction. To that end, we also introduce an extension to the existing ShapeStacks dataset which will be made publicly available. (2) A framework for interpreting, suppressing or promoting extraction of features specific to a secondary task unifying existing approaches from interpretability, auxiliary and adversarial learning. While we frame this work in the context of intuitive physics, questions regarding model interpretability and, consequently, systematic, targeted model adaptation find applicability in all domains of deep learning. For a study of two MNIST toy problems with neural stethoscopes, please see Appendix C.

## 2 RELATED WORK

This work touches on two fields within deep learning: intuitive physics, specifically stability prediction, and targeted model adaptation/interpretation.

**Stability Prediction** Vision-based stability prediction of block towers has become a popular task for teaching and showcasing physical understanding of algorithms. Approaches include end-to-end deep learning algorithms (Lerer et al., 2016; Groth et al., 2018) as well as pipelines using computer vision to create an abstract state representation which can then be used by a physics simulator (Furrer et al., 2017; Wu et al., 2017). Furrer et al. (2017) achieve impressive results with a pipeline approach to robotic stone stacking. Interestingly, on a publicly available real-world dataset of block tower images (Lerer et al., 2016), the state-of-the-art is shared between an end-to-end learning approach (Groth et al., 2018) and a pipeline method using a physics simulator (Wu et al., 2017). While being

much easier to implement, end-to-end approaches have the downside of being significantly harder to interpret. Interpretability brings at least two advantages: (1) Trust by understanding the model's reasoning and therefore also its potential failure cases. (2) Identification of potentials to improve the model. Both Lerer et al. (2016) and Groth et al. (2018) conduct occlusion-based attention analyses. Groth et al. (2018) find that the focus of the algorithm's attention lies within a bounding box around the stability violation in 80% of the cases. While encouraging, conclusions which can be drawn regarding the network's understanding of physical principles are limited. Moreover, Selvaraju et al. (2017) shows that attention analyses can be misleading: The Grad-CAM visualisation does not change even for artificially crafted adversarial examples which maximally confuse the classifier.

**Neural Stethoscopes** The notion of passively interrogating and actively influencing feature representations in hidden layers of neural networks connects disparate fields including interpretability, auxiliary losses and multitask learning as well as adversarial training. We note that much of the required machinery for neural stethoscopes already exists in a number of sub-domains of deep learning. Work on auxiliary objectives (Jaderberg et al., 2016) as well as multi-task learning (*e.g.* Caruana (1995; 1998)) commonly utilises dedicated modules with losses targeted towards a variety of tasks in order to optimise a representation shared across tasks. Based on this notion both *deep supervision* (Wang et al., 2015) and *linear classifier probes* (Alain & Bengio, 2016) reinforce the original loss signal at various levels throughout a network stack. Although their work is restricted to reinforcing the learning signal via the same loss applied to the global network (Alain & Bengio, 2016) in particular demonstrate that the accessibility of information at a given layer can be determined - and promoted - by formulating a loss applied locally to that layer.

Conversely, in order to encourage representations invariant to components of the input signal, regularisation techniques are commonly utilised (*e.g.* Srivastava et al. (2014)). To obtain invariance with respect to known systematic changes between training and deployment data, Ganin et al. (2016); Wulfmeier et al. (2017) propose methods for domain adaptation. In order to prevent a model fitting to a specific nuisance factor, Louizos et al. (2015) minimise the Maximum Mean Discrepancy between conditional distributions with different values of a binary nuisance variable in the conditioning set. This method is limited to discrete nuisance variables and its computational cost scales exponentially with the number of states. Xie et al. (2017) address both issues via adversarial training, optimising an encoding to confuse an additional discriminator, which aims to determine the values of nuisance parameters. This approach assumes that the nuisance variable is known and is an input to the main model during training and deployment. Louppe et al. (2017) follow a similar approach, applying the discriminator to the output of the main model instead of its intermediate representations.

## 3  NEURAL STETHOSCOPES

We use stethoscopes to analyse and influence the learning process on the task of stability prediction, but present it in the following as a general framework which can be applied to any set of tasks.

In supervised deep learning, we typically look for a function $f_\theta : X \to Y$ with parameters $\theta$ that maps an input $x \in X$ to its target $y \in Y$. Often the function internally computes one or more intermediate representations $z \in Z$ of the data. In this case, we rewrite $f_\theta$ as the composition of the encoder $h_\theta^{\text{enc}} : X \to Z$, which maps the input to the corresponding features $z \in Z$, and the decoder $h_\theta^{\text{dec}} : Z \to Y$, which maps features to the output. In this work, we consider only classification tasks so that $Y$ is a finite set of labels (or a simplex in the case of a probabilistic output), but our approach generalises directly also to regression tasks.

The information present in the input $x$ might support multiple different tasks. By introducing a supplementary task into our training framework we hope to improve the performance of our network at the primary task. However, the impact of the supplementary task on the primary task is often difficult to determine, and in some cases can even detriment performance on the primary task. To this end, we propose neural stethoscopes, which allow us to interrogate the relationships between primary and supplementary tasks. In addition, the framework provides a tool to improve the network's performance at the primary task in light of both beneficial and detrimental supplementary tasks through the introduction of *auxiliary* and *adversarial* losses respectively promoting or suppressing extraction of features related to the supplemental task by the encoder $h_\theta^{\text{enc}}$.

Let the stethoscope be defined as an arbitrary function $h_\psi^{\text{s}} : Z \to S$ with parameters $\psi$. Crucially, $h_\psi^{\text{s}}$ is trained on a supplemental task with targets $s \in S$ and not for the main objective. We define two

loss functions: $\mathcal{L}_y(\theta)$, which measures the discrepancy between predictions $f_\theta$ and the true task $y$ and $\mathcal{L}_s(\theta, \psi)$, which measures the performance on the supplemental task.

The weights of the stethoscope are updated as $-\Delta\psi \propto \nabla_\psi \mathcal{L}_s(\theta, \psi)$ to minimise $\mathcal{L}_s(\theta, \psi)$ and the weights of the main network as $-\Delta\theta \propto \nabla_\theta \mathcal{L}_{y,s}(\theta, \psi)$ to minimise the energy

$$\mathcal{L}_{y,s}(\theta, \psi) = \mathcal{L}_y(\theta) + \lambda \cdot \mathcal{L}_s(\theta, \psi). \tag{1}$$

By choosing different values for the constant $\lambda$ we obtain three very different use cases:

**Analytic Stethoscope ($\lambda = 0$)** Here, the gradients of the stethoscope, which acts as a passive observer, are not used to alter the main model. This setup can be used to interrogate learned feature representations: if the stethoscope predictions are accurate, the features can be used to solve the task.

**Auxiliary Stethoscope ($\lambda > 0$)** The encoder is trained with respect to the stethoscope objective, hence enforcing correlation between main network and supplemental task. This setup is related to learning with auxiliary tasks, and helpful if we expect the two tasks to be beneficially related.

**Adversarial Stethoscope ($\lambda < 0$)** By setting $\lambda < 0$, we train the encoder to maximise the stethoscope loss (which the stethoscope *still* tries to minimise), thus encouraging independence between main network and supplemental tasks. This is effectively an adversarial training framework and is useful if features required to solve the stethoscope task are a detrimental nuisance factor.

For the analytic stethoscope, to fairly compare the accessibility of information with respect to a certain task in different feature representations, we set two criteria: (1) The capacity of the stethoscope architecture has to be constant regardless of the dimensions of its input. (2) The stethoscope has to be able to access each neuron of the input separately. We guarantee this by fully connecting the input with the first layer of the stethoscope using a sparse matrix. This matrix has a constant number of non-zero entries (criterion 1) and connects every input unit as well as every output unit at least once (criterion 2). For a more detailed description, see Appendix A.

In auxiliary and adversarial mode, we attach the stethoscope to the main network's last layer before the logits in a fully connected manner. This setup proved to have the highest impact on the learning process of the main network. The stethoscope itself is implemented as a two-layer perceptron with ReLU activation and trained with sigmoid or softmax cross-entropy loss on its task $\mathcal{S}$.

For numerical stability, the loss of the encoder in the adversarial setting is rewritten as

$$\tilde{\mathcal{L}}_{y,s}(\theta, \psi) = \mathcal{L}_y(\theta) + |\lambda| \cdot \mathcal{L}_{\bar{s}}(\theta, \psi) \tag{2}$$

where $\mathcal{L}_{\bar{s}}(\theta, \psi)$ is the stethoscope loss with flipped labels. The objective is similar to the confusion loss formulation utilised in GANs to avoid vanishing gradients when the discriminator's performance is high (Goodfellow, 2016).

## 4 VISION-BASED STABILITY PREDICTION OF BLOCK TOWERS

Previous work has shown that neural networks are highly capable of learning physical tasks such as stability prediction. However, unlike approaches using physics simulators (Furrer et al., 2017; Wu et al., 2017), with pure-learning based approaches, it is hard to assess what reasoning they follow and whether they gain a sound understanding of the physical principles or whether they learn to take short-cuts following visual cues based on correlations in the training data. Occlusion-based attention analyses are a first step in this direction, but the insights gained from this are limited (Lerer et al., 2016; Groth et al., 2018). In this section, we follow the state-of-art approach on visual stability prediction of block towers and examine as well as influence its learning behaviour. We introduce a variation of the ShapeStacks dataset from Groth et al. (2018) which is particularly suited to study the dependence of network predictions on visual cues. We then examine how suppressing or promoting the extraction of certain features influences the performance of the network using neural stethoscopes.

**Dataset** As shown in Groth et al. (2018), a single-stranded tower of blocks is stable if, and only if, at every interface between two blocks the centre of mass of the entire tower above is supported by the convex hull of the contact area. If a tower satisfies this criterion, *i.e.*, it does not collapse, we call it *globally stable*. To be able to quantitatively assess how much the algorithm follows visual cues, we introduce a second label: We call a tower *locally stable* if, and only if, at every interface between two blocks, the centre of mass of the block immediately above is supported by the convex

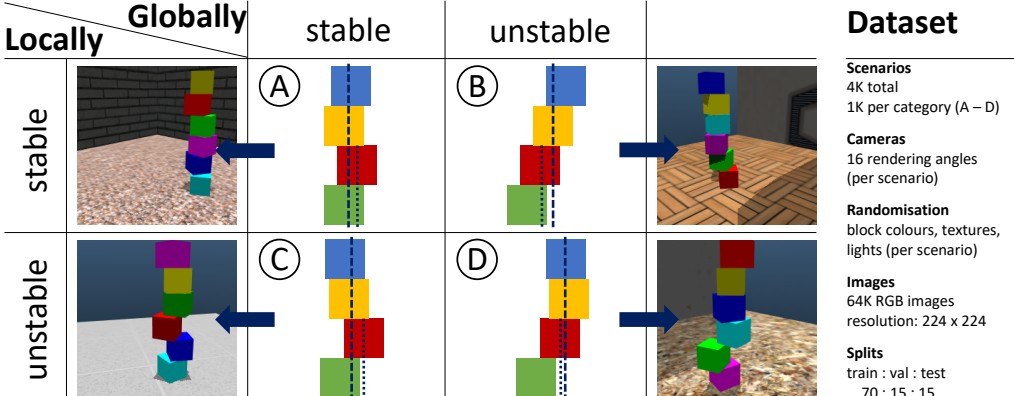

Figure 2: We have four qualitative scenarios: (A) Globally stable towers which also do not exhibit any local stability violation. (B) Towers without any local instability which are globally unstable because of the skew. (C) Towers in which one local stability violation is counterbalanced by blocks above. (D) Globally unstable towers where the site of local and global violation is perfectly correlated. The dashed line shows the projection of the cumulative centre of mass of the upper tower (red, yellow and blue block) whereas the dotted line depicts the projection of the local centre of mass of the red block. A tower is globally stable, if and only if the global centre of mass is always supported whereas the individual local centre of masses are not indicative of global structure stability. Global and local centre of masses for the green, yellow and blue block have been omitted for clarity of presentation.

hull of the contact area. Intuitively, this measure describes, if taken on its own without any blocks above, each block would be stable. We associate binary prediction tasks $y_G$ and $y_L$ to respective global and local stability where label $y = 0$ indicates *stability* and $y = 1$ *instability*. Global and local instability are neither mutually necessary nor sufficient, but can easily be confused visually which is demonstrated by our experimental results. Based on the two factors of local and global stability, we create a simulated dataset[1] with 4,000 block tower scenarios divided into four qualitative categories (cf. Figure 2). The dataset is divided into an *easy* subset, where local and global stability are always positively correlated, and a *hard* subset, where this correlation is always negative. The dataset will be made available online.

**Model** We choose the Inception-v4 network (Szegedy et al., 2017) as it yields state-of-the-art performance on stability prediction (Groth et al., 2018). The model is trained in a supervised setting using example tuples $(x, y_G, y_L)$ consisting of an image $x$ and its global and local stability labels $y_G$ and $y_L$. Classification losses use sigmoid cross entropy. We use the RMSProp optimiser (Tieleman & Hinton, 2012) throughout all our experiments.[2]

**Local Stability as a Visual Cue** Based on the four categories of scenarios described in Figure 2, we conduct an initial set of experiments to gauge the influence of local stability on the network predictions. If local stability had, as it would be physically correct, no influence on the network's prediction of global stability, the performance of the network should be equal for *easy* and *hard* scenarios, regardless of the training split. However, Figure 3 shows a strong influence of local stability on the prediction performance. When trained on the entire, balanced data set, the error rate is three times higher for hard than for easy scenarios (6% vs. 2%). When trained on easy scenarios only, the error rate even differs by a factor of 13. Trained on hard scenarios only, the average performance across all four categories is on the level of random chance (51%), indicating that negatively correlated local and global stability imposes a much harder challenge on the network.

---

[1]We use the MuJoCo physics engine (Todorov et al., 2012) for rendering and stability checking.

[2]We use the Tensorflow (Abadi et al., 2016) implementation of RMSProp without momentum, gradient history decay of 0.9, epsilon of 1.0, a learning rate of 0.045 and a learning rate decay of 0.975 after every epoch.

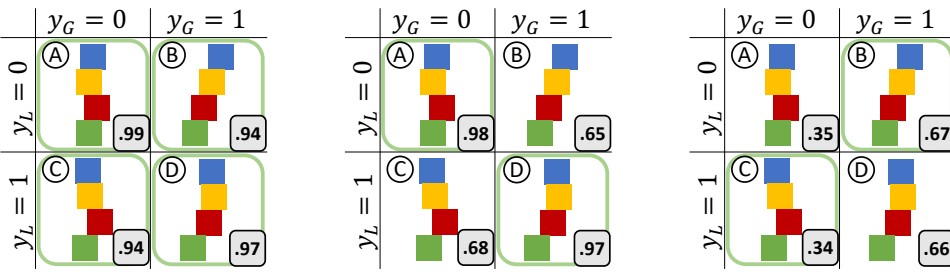

(a) Trained on All: $\varnothing_{acc} = 0.96$    (b) Trained on Easy: $\varnothing_{acc} = 0.82$    (c) Trained on Hard: $\varnothing_{acc} = 0.51$

Figure 3: The influence of local instability on global stability prediction. In setup (a) we train on all 4 tower categories (indicated by green frames). Global stability prediction accuracies on per-category test splits are reported in the bottom right grey boxes. In (b) we train solely on easy scenarios (A & D) where global and local stability are positively correlated. In (c) we only present hard scenarios during training featuring a negative correlation between global and local stability. The performance differences clearly show that local stability influences the network's prediction for global stability.

## 5   USING NEURAL STETHOSCOPES TO GUIDE THE LEARNING PROCESS

After demonstrating the influence of local stability on the task of global stability prediction we turn our attention to the use of neural stethoscopes to quantify and actively mitigate this influence.

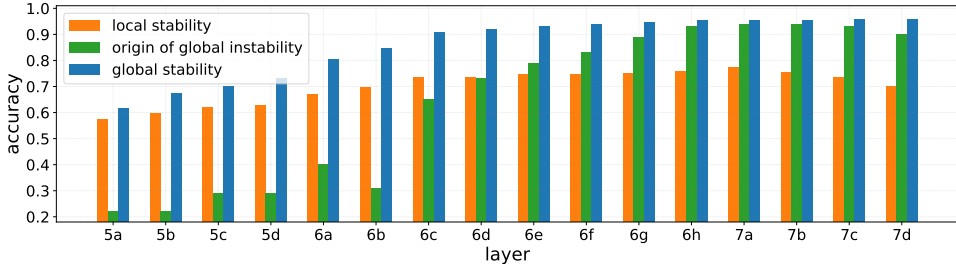

Figure 4: Analysis of the prediction performance throughout the Inception-v4 network trained on predicting global stability. We report average performances on balanced test data after 50 epochs of stethoscope training. All stethoscopes have been attached to the respective activation tensors[3] with sparse connection matrices as described in Section 3.

**Task Relationship Analysis**   We seek to quantify the correlation of features extracted for global stability prediction with the task of local instability detection. We train the main network on global stability prediction while attaching stethoscopes to multiple layers. The stethoscope is trained on three tasks: global stability prediction (binary), local stability (binary) and origin of global instability, particularly the interface at which this occurs (n-way one-hot). Figure 4 reveals that the network gradually builds up features which are conducive to both global and local stability prediction. However, the performance of local stability prediction peaks at the activation tensor after layer 7a whereas the performance for global stability prediction (both binary and n-way) keeps improving. This is in line with the governing physical principles of the scenarios and yields the conjecture that the information about local stability can be considered a *nuisance factor* whereas information about the global site of stability violation can serve as a *complementary factor* for the main task.

**Promotion of Complementary Information**   We now test the hypothesis that fine-grained labels of instability locations help the main network to grasp the correct physical concepts. To that end,

---

[3]The stethoscopes are connected to outputs of inception and reduction modules of the Inception v4 network, cf. Szegedy et al. (2017). Note that this analysis is specific to this architecture and could yield very different results, *e.g.*, for networks with residual connections.

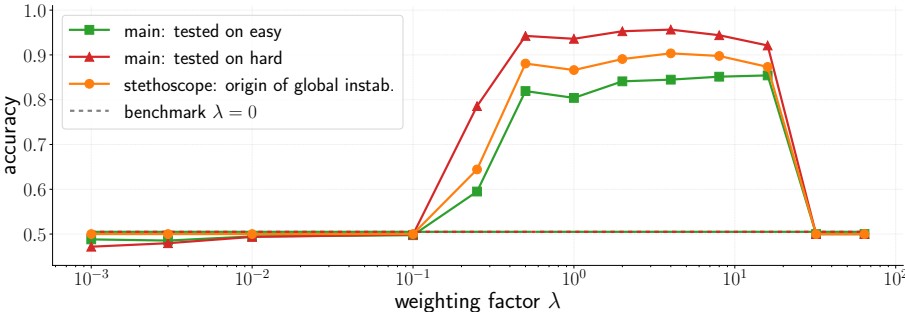

Figure 5: Performance gains by promoting complementary feature extraction with auxiliary stethoscopes. The main network is trained on binary global stability labels while the stethoscope is trained on more fine grained labels - origin of global stability (n-way). The network was trained on *hard* scenarios only but evaluated on all. The dashed lines represent baselines for $\lambda = 0$.

we consider the setup from Figure 3c where the training data only consists of *hard* scenarios with a baseline performance of 51%. The main network is trained on global stability while the stethoscope is trained on predicting the origin of global instability, namely the interface at which the instability occurs. Figure 5 shows that auxiliary training substantially improves the performance for weighting parameters $\lambda \in [0.5, 16]$. However, for very small values of $\lambda$, the contribution of the additional loss term is too small while for large values, performance deteriorates to the level of random chance as a result of the primary task being far out-weighted by the auxiliary task.

**Suppression of Nuisance Information**    Results from Figure 3 indicate that the network might use local stability as a visual cue to make biased assumptions about global stability. We now investigate whether it is possible to debias the network by forcing it to pay less attention to local stability. To that end, we focus on the scenario shown in Figure 3b, where we only train the network on global stability labels for *easy* scenarios. As shown in Figure 3b, the performance of the network suffers significantly when tested on *hard* scenarios where local and global stability labels are inversely correlated.

The hypothesis is that forcing the network not to focus on local stability weakens this bias. In Figure 6, we use active stethoscopes ($\lambda \neq 0$) to test this hypothesis. We train a stethoscope on local stability on labels of all categories (in a hypothetical scenario where local labels are easier to obtain than global labels) and use both the adversarial and the auxiliary setup in order to test the influence of suppressing and promoting accessibility of information relevant for local stability in the encoded representation, respectively. In Figure 6, the results of both adversarial and auxiliary training are compared to the baseline of $\lambda = 0$, which is equivalent to the analytic stethoscope setup.

Figure 6a shows that adversarial training does indeed partly remove the bias and significantly increases the performance of the main network on *hard* scenarios while maintaining its high accuracy on *easy* scenarios. The bias is removed more and more with increasing magnitude of the weighting factor $\lambda$ up to a point where further increasing $\lambda$ jeopardises the performance on the main task as the encoder puts more focus on confusing the stethoscope than on the main task (in our experiments this happens at $\lambda \approx 10^1$). Interestingly, the correlation of local stability and global stability prediction rises at this point as for pushing the stethoscope below 50% (random guessing), the main network has to extract information about local stability. Naturally, the performance of the stethoscope continuously decreases with increasing $\lambda$ as the encoder puts more and more focus on confusing the stethoscope.

This scenario could also be seen from the perspective of feeding additional information into the network, which could profit from more diverse training data. However, Figure 6b shows that naively using an auxiliary setup to train the network on local stability worsens the bias. With increasing $\lambda$ and increasing performance of the stethoscope, the network slightly improves its performance on *easy* scenarios while accuracy deteriorates on *hard* scenarios. These observations are readily explained: local and global stability are perfectly correlated in the *easy* scenarios, *i.e.*, the (global) training data. The network is essentially free to chose whether to select local or global features when predicting global stability. Auxiliary training on local stability further shifts the focus to local features. When tested on *hard* scenarios, where local and global stability are inversely correlated, the network will therefore perform worse when it has learned to rely on local features.

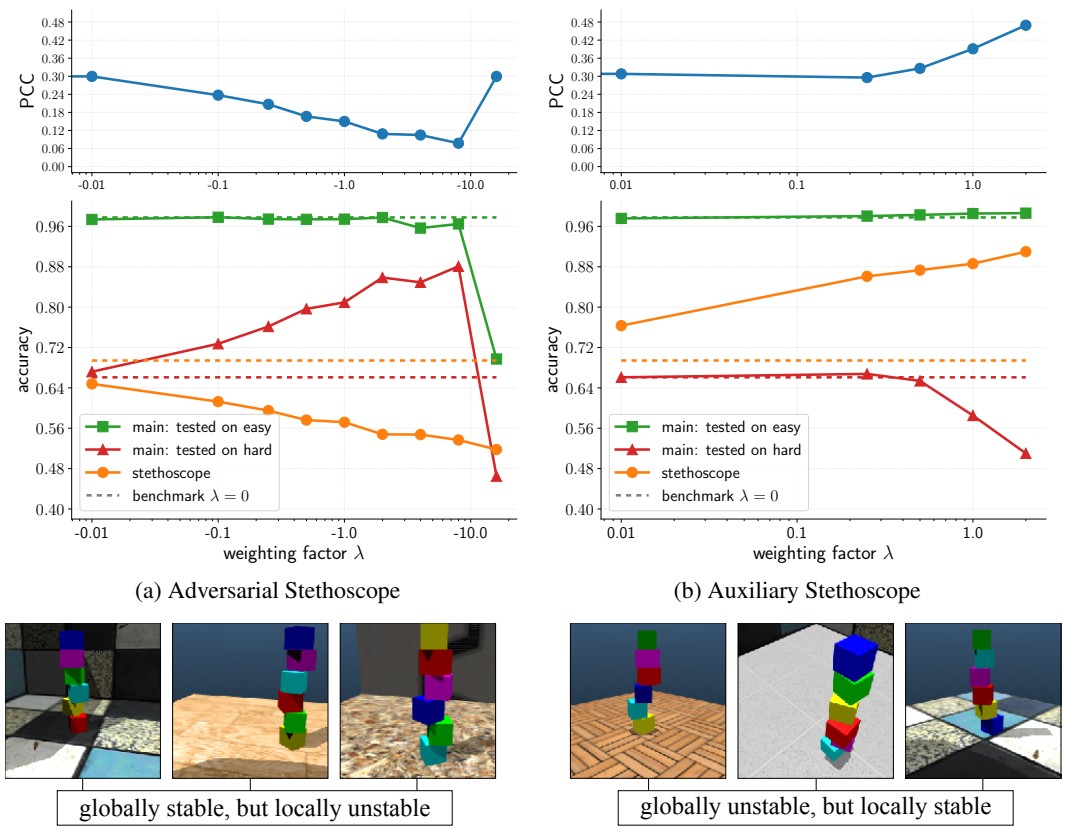

(a) Adversarial Stethoscope        (b) Auxiliary Stethoscope

globally stable, but locally unstable       globally unstable, but locally stable

(c) Cases where the algorithm predicted incorrectly without and correctly with adversarial training ($\lambda = -8.0$).

Figure 6: Successful debiasing by suppressing a nuisance factor with adversarial training while auxiliary training worsens the bias. The main network is trained on global stability while the supplementary task is to predict presence of local instabilities. Training data for global stability only comprises *easy* scenarios while training data for the supplementary task comprises both *easy* and *hard* scenarios. **(a)** & **(b)** Green squares and red triangles show accuracies of the main network when predicting global stability of block towers for *easy* scenarios and *hard* scenarios, respectively. Orange circles depict the performance of the stethoscope on the task of local stability. Blue circles represent the Pearson Correlation Coefficient of predicted global stability and ground truth local stability which gives an indication of how much the network follows visual cues. **(c)** Images show *hard* scenarios which the algorithm classified correctly only when adversarial training was used.

## 6 DISCUSSION

We study the state-of-the-art approach for stability prediction of block towers and test its physical understanding. To that end, we create a new dataset and introduce the framework of *neural stethoscopes* unifying multiple threads of work in machine learning related to analytic, auxiliary and adversarial probing of neural networks. The analytic application of stethoscopes allows measuring relationships between different prediction tasks. We show that the network trained on stability prediction also obtains a more fine-grained physical understanding of the scene (origin of instability) but at the same time is susceptible to potentially misleading visual cues (*i.e.*, local stability). In addition to the analysis, the auxiliary and adversarial modes of the stethoscopes are used to support beneficial complementary information (origin of instability) and suppress harmful nuisance information (visual cues) without changing the network architecture of the main predictor. This yields substantial performance gains in unfavourable training conditions where data is biased or labels are partially unavailable. We encourage the use of neural stethoscopes for other application scenarios in the future as a general tool to analyse task relationships and suppress or promote extraction of specific features. This can be done by collecting additional labels or using existing multi-modal data.

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

## A    CONNECTION OF STETHOSCOPE TO MAIN NETWORK

This is a detailed explanation of how to connect the first layer of the stethoscope with any layer of the main network as described in Section 3. The goal is to guarantee equal capacity of the stethoscope regardless of the dimensions of the layer it is connected to in order to obtain comparable results. Let's call the flattened feature vector from the main network $\vec{x}$. In the following we describe how to compute the sparse matrix $\mathbf{M}$ which reduces the full vector $\vec{x}$ to a smaller vector $\vec{x'}$ while adding biases:

$$\vec{x'} = \mathbf{M} \cdot \vec{x} + \vec{b} \tag{3}$$

$\mathbf{M}$ is a sparse matrix with a constant number of non-zero entries ($n\_non\_zero$) which are trainable. The hyperparameter $n\_non\_zero$, which determines the capacity of this additional linear layer, is chosen as an integer multiple of the length of $\vec{x'}$ and has to be greater or equal than the number of features in any of the layers of the main network which the stethoscope is attached to. The matrix $\mathbf{M}$ is generated in a way so that it fulfils the following two guarantees:

- Each input is connected at least once. *I.e.*, no column only contains zeroes.
- Each output is connected an equal amount of times. *I.e.*, all rows have the same number of non-zero entries.

## B    APPENDIX: IS INFORMATION DISCARDED IN ADVERSARIAL TRAINING?

Figure 6a shows that adversarial training increases performance given a biased training dataset. While hypothetically the network could learn to completely discard information about local stability when trying to confuse the adversarial stethoscope, it could also simply reduce its accessibility.

In an additional experiment (Figure 7), where the main network (blue line) is fixed after adversarial training for 120 epochs (grey dashed line), the performance of the stethoscope on local stability (orange) is fully recovered to the level of the baseline for a purely analytic stethoscope (dashed orange line). We therefore argue that the information is not removed, but instead made less accessible to the stethoscope by constantly shifting the representation as explained in (Mescheder et al., 2017). However, the resulting performance increase for the main task indicates that this behaviour also

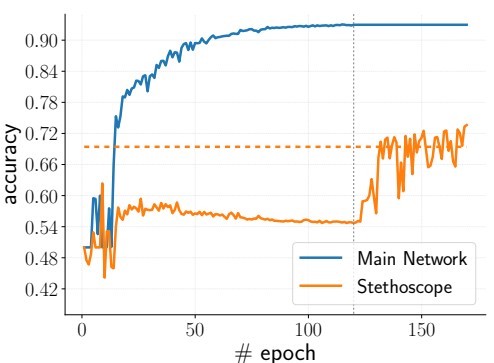

Figure 7: Stethoscope recovery after freezing the adversarially trained main network.

makes information related to local stability less accessible to the decoder of the main network. Hence, although the adversarial setup does not reach final, stable equilibria (*cf.* (Mescheder et al., 2017)), it decreases the dependence of the decoder on features particular to local stability. Hence, further improvements for stabilising GAN training could additionally improve performance in adversarial stethoscope use-cases (Mescheder et al., 2017; Goodfellow, 2016; Arjovsky et al., 2017).

## C    APPENDIX: SUPPRESSING NUISANCE INFORMATION IN MNIST

This section introduces two intuitive examples and serves to illustrate the challenge with biased training datasets. We conduct two toy experiments built on the MNIST dataset to demonstrate the issue of varying correlation between nuisance parameters and ground truth labels in training and test datasets and to demonstrate the use of an adversarial stethoscope to suppress this information and mitigate overfitting. Both experiments build on the provision of additional *hints* as network input. In the first experiment, the hint is a separately handled input in the form of a one-hot vector whereas in the second experiment, the hint is directly incorporated in the pixel space of the MNIST images.

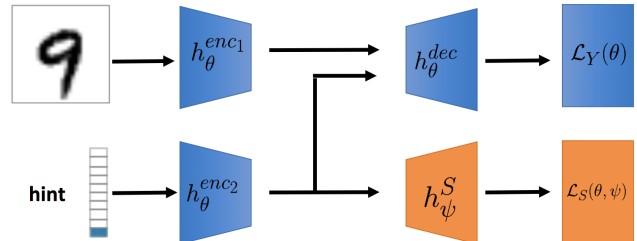

(a) First Toy Example: Hint as Separate One-Hot Vector

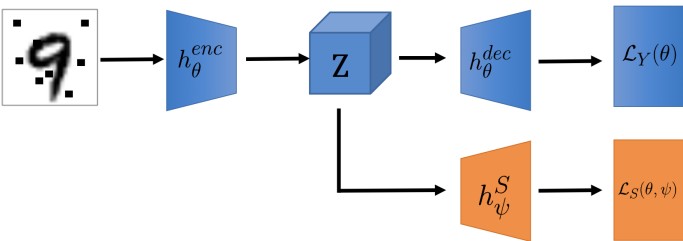

(b) Second Toy Example: Hint as Pixel Encoding

Figure 8: Setup of the toy experiments using the MNIST dataset with artificial hints. In (a), the hints are fed to the main network via a separate input and both input streams are encoded separately by neural networks. The stethoscope only sees the output of the hint encoder $h_\theta^{enc_2}$ while the decoder $h_\theta^{dec}$, which is trained on digit classification, sees the output of both encoders. In (b), the hints are incorporated as pixel modifications into the main images. The encoder $h_\theta^{enc}$ transforms the input into a latent representation Z which is then accessed by both the stethoscope $h_S^\psi$ and the decoder $h_\theta^{dec}$.

In both experiments, the network is trained on the loss as defined in Equation (1) and the gradients are used to update the weights of the different parts of the network (encoder(s), decoder, stethoscope) as defined in Section 3. Hence, $\lambda < 0$ denotes adversarial training and $\lambda > 0$ auxiliary training.

**Hint as Separate One-Hot Vector** In the first example, the hint labels have the same format as the true labels of the input image and we run experiments with varying correlation between hint and true label in the training set. Given high correlation, the network can learn an identity transformation between hint and output and ignore the image. Now, at test time, the hint is completely independent of the true label and hence contains no useful information. The experiment mimics challenges with biased datasets as the supplemental task $s \in \mathcal{S}$ (see Section 3) has labels $\boldsymbol{y_s}$ which are correlated to the labels $\boldsymbol{y}$ of the main objective during training but not at test time ($p_{train}(\boldsymbol{y_s}|\boldsymbol{y}) \neq p_{test}(\boldsymbol{y_s}|\boldsymbol{y})$). Hence, an approximation of the correlations in the training data will not generalise to data during deployment.

To that end, we introduce a parameter $q_h$ denoting the quality of the hints (i.e. their correlation with the true labels) during training time where for $q_h = 1.0$, the hints are always correct (during training), for $q_h = 0.5$, the hints are correct for $50\%$ of the training examples and so on. Varying the hint quality enables us to investigate biases of increasing strength.

In this setup (Figure 8a), both input streams are separated via independent encoders. Let $h_\theta^{enc_1}$ and $h_\theta^{enc_2}$ respectively denote image and hint encoders. The stethoscope only accesses the hint encoding in this example, which simplifies its task and enables us to demonstrate the effect of active stethoscopes without affecting the image encoding. The hint encoder multiplies the one-hot hint vector with a scalar weight. Hence, a weight close to zero would effectively hide the hint information from the main network, explicitly making the task of the stethoscope $h_S^\psi$ more difficult. The image encoder consists of two convolutional layers followed by two fully connected layers with 256 units where each layer is followed by leaky ReLU activations. The decoder $h_\theta^{dec}$, which acts as a digit classifier, is comprised of a single fully connected layer and receives the image encoding concatenated with the encoded hint vector as input.

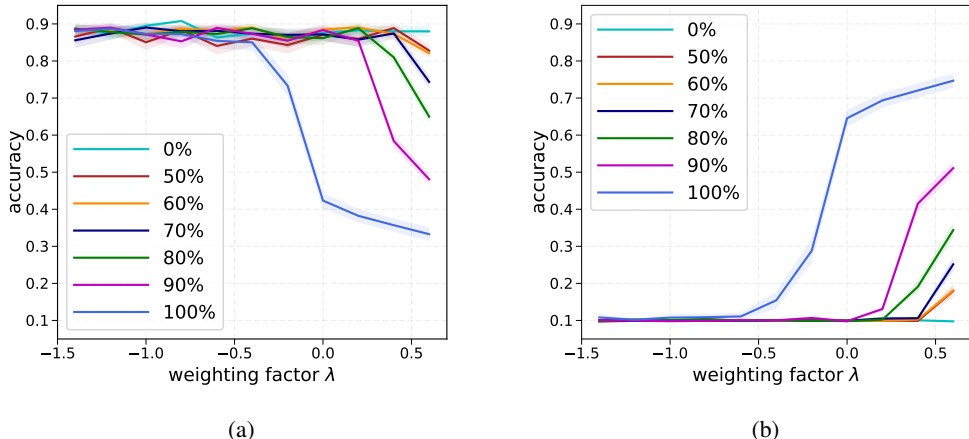

|     |     |
| --- | --- |
| (a) | (b) |

Figure 9: Influence of adversarial training on classifier performance for toy experiment variant 1 described in Figure 8a. Each curve represents the results for different hint qualities $h_q$. The bold lines indicate the mean of 100 runs and the shaded areas mark the student-t $1 \sigma$ confidence intervals. In (a) the network is evaluated against the ground truth. In (b), the output of the network is compared to the hint labels during test time. Hence, a high accuracy in (b) shows strong overfitting while a high accuracy in (a) shows that the network learned the actual task of classifying digits from images.

With adversarial training of the stethoscope, the encoder learns to suppress information about the hints forcing the classifier to purely rely on image data. As can be observed in Figure 9b, for perfect hints ($q_h = 1.0$) and no adversarial training, the decoder is highly accurate in predicting the hint (instead of the actual label), suggesting that the classifier is effectively independent of the input image and purely relies on the hint. With adversarial training, however, we can indeed force the encoder to hide the hint, therefore forcing it to learn to classify digits from images. As expected, with increasing hint quality, we need to increase the weight of adversarial training so as to discourage the encoder from relying on the hint.

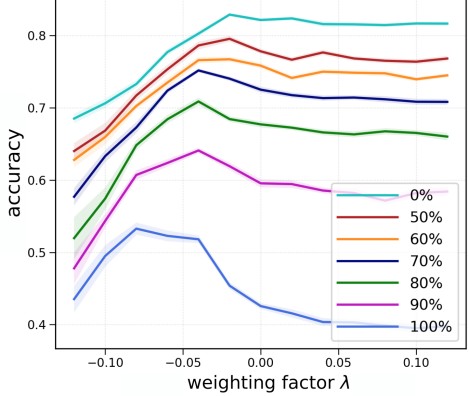

Figure 10: Influence of adversarial training on classifier performance for the second MNIST experiment. The adversarial training is less effective as both the stethoscope and the main network directly access a single encoding. High adversarial weights can strongly degrade performance on the main task as a trivial solution would be to suppress all information about the input. Each curve represents the results for different hint qualities $h_q$. The bold lines indicate the mean of 20 runs and the shaded areas mark the student-t $1 \sigma$ confidence intervals.

**Hint as Pixel Encoding**   In the second variant of the MNIST experiment, hints are provided at training time as a set of high intensity pixels in the input images as opposed to the explicit concatenation in the previous example. This better reflects the typical scenario where the goal is

to disentangle the nuisance hints from the informative information related to the main task. Here, the main network is a two-layer multi-layer perceptron (MLP) with the stethoscope attached after the first hidden layer. The simpler architecture (compared to the one used in the first toy problem) was chosen in order to lower the performance on vanilla MNIST classification in order to see clearer effects. Note that in this setup, giving the same labels to the hints as for the main task in the training scenario makes the two tasks indistinguishable (in particular for perfect hint quality $h_q = 1.0$). We therefore introduce a higher number of hint classes with 100 different hints, each of them related to a single digit in a many-to-one mapping, such that theoretical suppression of hint information is possible without degrading main performance.

Figure 10 shows the digit classification accuracy for the main network where the stethoscope is applied with weights ranging in both the positive and negative directions to reflect auxiliary and adversarial use of the stethoscope respectively. When using a positive stethoscope loss, which encourages $h_\theta^{enc}$ to focus more on the artificial hints over the actual digit features, the test accuracy degrades. This is expected for this toy example as the hints have zero correlation with the digit labels at test time. In the negative case, $h_\theta^{enc}$ serves as an adversary to the stethoscope effectively benefiting the main objectives by removing the misleading hint information leading to an improved test accuracy. However, as the magnitude of the adversarial strength increases the gains quickly erode as the encoder begins to suppress information relevant for the main task.

