# OpenReview forum: "Guiding Physical Intuition with Neural Stethoscopes"
_ICLR.cc/2019/Conference_

### Official Review · AnonReviewer1 · 2018-11-02
**nice solid paper**

**Rating:** 7
**Confidence:** 3

**Review:**

This paper is about using "neural stethoscopes", small complementary neural networks that are added to a main network which with their auxilary loss functions can measure suitability of features or guide the learning process. The idea is incremental to multi-task learning and enable, in a single framework, to validate intermediate features for additional related tasks. Moreover it can promote or suppress the correlation of such features to the tasks related to the main one. The framework is applied to the task of visual stability prediction of block towers. The paper builds upon Groth et al. 2018, adding the concept of local stability as correlated secondary task, used with the proposed neural stethoscopes. Experiments with an extension of ShapeStacks (Groth et al. 2018) dataset where the local stability is added to the global stability class, show that it is possibile increase the performance using the additional task. Moreover, it is shown that neural stethoscopes can suppress nuisance information when using a biased training dataset where the local and global stability are purposely inversely correlated.

Strengths:
+ A very nice paper, well written and easy to read. Figures are helpful and the structure is clear.
+ The concept of neural stethoscope is interesting and simplify the concepts behind multitask learning.
+ Experiments are convincing, interesting and there is some novelty in vision stability prediction.

Weaknesses:
- The novelty is limited related to multitask learning, thus it is an incremental paper.

---

> ### Author Response · Authors · 2018-11-09
> **Reply to Reviewer 1**
>
> Thank you for your review. We believe it captures the essence of what we are proposing and are delighted with the overall very positive assessment. As regards the weakness mentioned, we accept the characterisation of our work as incremental in the sense that it draws together a number of known techniques from areas such as multi-task and adversarial learning. We argue, however, that the contribution of our submission lies exactly in the unification of these approaches into the stethoscope framework, which lends itself to targeted representation analysis and modification. We showcase stethoscopes to provide insights into a particular application domain, stability prediction in intuitive physics, but believe that the methods presented here will provide a ready toolkit for researchers addressing a variety of challenges in network interpretability and (de)biasing.

---

### Official Review · AnonReviewer2 · 2018-11-03
**Preliminary report, the idea could be shaped into a better one**

**Rating:** 4
**Confidence:** 3

**Review:**

This paper combines the global and local stability prediction and tries to get interpretable results using the stethoscope design, which is actually a weighted subbranch for the main branch. There are several concerns regarding the proposed framework.

1) How to choose \lambda? A better design could be a learnable \lambda. Instead of just one scalar value, it could be better to learn a map of \lambdas, which indicates the distribution of local stability and how it is related to global stability. The visualization of the \lambda map might be more interpretable for understanding the stability prediction.

2) The global stability prediction does not have a consistent correlation with the local stability prediction, as shown by the easy and hard examples. This complex relationship will confuse the network during the training. That is, the current design hasn't well considered the local and global stability relation, but just simply sum them up. This is hard to provide a meaningful interpretation of the task.

---

> ### Author Response · Authors · 2018-11-09
> **Reply to Reviewer 2**
>
> Thank you for your review and we would be delighted to address your concerns, but do require some clarifications.
>
> While a learnable lambda could be considered we would argue that the learning of this parameter beyond the grid-search applied in the submission is somewhat tangential to our primary contribution: a unified framework which lends itself to targeted representation analysis and modification.
>
> 1)
> The notion of a map of \lambdas sounds interesting. However, at present, it is not clear to us what this refers to as \lambda is a weighting on a loss term. Clarification would be much appreciated so we can fully engage with this point. As far as the existing approach is concerned, Figure 6 illustrates the influence of \lambda on the accuracy and correlation of global and local stability prediction.
>
> 2)
> The inconsistent correlations between the two tasks are exactly the scenarios where stethoscopes come into their own: testing positive and negative regimes of lambda (corresponding to auxiliary and adversarial training, respectively) reveals the interplay between the two tasks and potentially allows for de-biasing the algorithm as shown in Figure 6a. Therefore, in contrast to the design not considering these relationships, it explicitly addresses them.
>
> Could you please elaborate on the comment ’the current design […] simply sums them up’? The stethoscope module has its own trainable parameters and a separate loss function. Only the encoder shares weights between main and secondary task.

---

### Official Review · AnonReviewer3 · 2018-11-07
**A well-written paper with limited implication.**

**Rating:** 6
**Confidence:** 4

**Review:**

The paper focuses on the stability prediction task on the ShapeStacks dataset. Specifically, the paper creates a new extension to the dataset, and it proposes the use of "Neural Stethoscopes" framework to analyze deep neural nets' physical reasoning of local stability v.s. global stability. It is shown in the paper neural nets tend to be misled by local stability when the task is to predict global stability. Then the paper utilizes the proposed framework to de-bias the misleading correlation to achieve a state-of-the-art on the dataset.

The paper is very well-written and easy to follow. The main idea is simple and the experiments are detailed. Specifically on the task of stability prediction, it is quite interesting to know that neural nets can be misled by visual cues (local stability).

However, my concern is that the paper focuses only on a very specific application domain,  and an improvement over the niche dataset with much more supervision (from the extension) is not surprising at all. In the mean time, the notion of "Neural Stethoscopes" could be much more  generally applied. Without applications in other domains, it is not immediately clear what the paper's implication is.

---

> ### Author Response · Authors · 2018-11-09
> **Reply to Reviewer 3**
>
> Thank you for your review and the overall positive assessment. In particular, we are delighted that you see the potential of the stethoscope framework lending itself to much broader applications. We agree - particularly for applications regarding the interpretability of deep representations as well as the manipulation of biases contained therein. This is a key consideration in our belief that it makes for a valuable contribution to the community.
>
> We have opted, in this submission, to focus on our primary application domain, which is intuitive physics. In particular, we demonstrate the efficacy of neural stethoscopes in interpreting, promoting and suppressing specific information in the context of the complex interplay between visual clues and physical properties in stability prediction. Our work is primarily motivated by the question as to what extent neural networks learn about physical principles or whether they merely follow visual clues and how we can guide the learning process. We showcase the stethoscope framework here to that effect and, based on its application, provide some interesting and novel [according to Reviewer 1] insights into representations for visual stability prediction.
>
> As regards the implications of the paper, we would like to address this in the context of the reviewer’s comment that increased performance is not surprising given the additional supervision provided. Our submission argues that the manner in which this information is provided really does matter. Figure 6 addresses this point in that multi-task learning fails to leverage the potential of the additional training labels (and, indeed, leads to a detrimental effect, Fig 6b) whereas the stethoscope framework allows the specification of whether the information considered should be promoted or suppressed. This leads to the performance gains shown in Fig 6a.

---

### Meta-Review · Area_Chair1 · 2018-12-17
**Interesting idea for which the presented evaluation is too narrow**

**Confidence:** 4
**Recommendation:** Reject

**Metareview:**

This submission proposes an interesting new approach on how to evaluate what features are the most useful during training. The paper is interesting and the proposed approach has the potential to be deployed in many applications, however the work as currently presented is demonstrated in a very narrow domain (stability prediction), as noted by all reviewers. Authors are encouraged to provide stronger experimental validation over more domains to show that their approach can truly improve over existing multitask frameworks.